

# Identification of exosomal miRNAs associated with the anthracycline-induced liver injury in postoperative breast cancer patients by small RNA sequencing

Yue Zhang[1], Di Wang[2], Di Shen[1], Yang Luo[3] and Yi-Qun Che[1]

[1] Department of Clinical Laboratory, National Cancer Center/ National Clinical Research Center for Cancer/ Cancer Hospital, Chinese Academy of Medical Sciences and Peking Union Medical College, Beijing, P.R.China
[2] State Key Lab of Molecular Oncology, National Cancer Center/ National Clinical Research Center for Cancer/ Cancer Hospital, Chinese Academy of Medical Sciences and Peking Union Medical College, Beijing, P.R.China
[3] Department of Medical Oncology, National Cancer Center/ National Clinical Research Center for Cancer/ Cancer Hospital, Chinese Academy of Medical Sciences and Peking Union Medical College, Beijing, P.R.China

Corresponding authors
Yang Luo, luoyang@cicams.ac.cn
Yi-Qun Che, cyq@cicams.ac.cn

## ABSTRACT

**Background.** Anthracycline-induced liver injury (AILI) is one of the serious complications of anthracycline-based adjuvant chemotherapy for postoperative breast cancer patients. Exosomal miRNAs, as signaling molecules in intercellular communication, play the essential roles in drug-induced liver injury (DILI). However, the expression profiles of them in patients with AILI remains unknown.

**Methods.** Seven post-chemotherapy patients were recruited in this study. After isolated plasma-derived exosomes, small RNA sequencing revealed exosomal miRNA profiles and differentially expressed miRNAs (DE-miRNAs) were identified between the liver injury group and non-liver injury group. miRTarBase and miRDB were used to predict the potential target genes of DE-miRNAs. DILI-related genes were downloaded from the CTD Database. The intersection of predicted genes and DILI-related genes were identified as the AILI-related target genes of the DE-miRNAs. GO annotation and KEGG pathway enrichment analysis were performed by the DAVID database. Furthermore, the protein-protein interaction (PPI) network was established by the STRING database and essential exosomal miRNAs were identified via Cytoscape software.

**Results.** A total of 30 DE-miRNAs and 79 AILI-related target genes were identified. AILI-related target genes of the DE-miRNAs are significantly enriched in NOD-like receptor signaling pathway, the HIF-1 signaling pathway, and the FoxO signaling pathway. Then, the hub genes were screened and we discovered that IL-6 and SOD2 are the most critical genes that may be involved in the development of AILI through the activation of immune response and the occurrence of oxidative stress, respectively. In addition, we found that miR-1-3p could potentially regulate most of the hub genes in the miRNA-hub gene network.

**Conclusion.** We explored the potential functions of DE-miRNAs and suggested exosomal miR-1-3p might be the essential exosomal miRNA in the pathogenesis of AILI. Moreover, our study provided an experimental basis for experimental verification to reveal the actual function and mechanism of miRNAs in AILI.

## INTRODUCTION

The liver is the largest internal organ responsible for the selective uptake, metabolism, and excretion of endogenous and exogenous compounds, including drugs. The activity of uptake transporters that facilitate the accumulation of drugs in hepatocytes makes this organ susceptible to drug-induced liver injury (DILI) (*Kolarić et al., 2019*). The true incidence of DILI is difficult to determine due to its lack of standardized diagnostic criteria, delay in diagnosis and unpredictable nature, but the population incidence is likely on the rise. DILI represents a serious clinical problem and potentially fatal course (*Hayashi & Chalasani, 2015*; *Cano-Paniagua et al., 2019*).

Breast cancer is the most frequently diagnosed cancer and the leading cause of cancer death in females worldwide (*Siegel, Miller & Jemal, 2019*). Treatment with adjuvant chemotherapy is recommended for women with resected node-positive or high-risk node-negative breast cancer, and an anthracycline-based regimen is often included (*Waks & Winer, 2019*). Anthracycline, mainly including doxorubicin and epirubicin, is a class of antibiotics derived from the Streptomyces bacteria and has pleiotropic effects including free radical formation, topoisomerase II inhibition and altered mitochondrial function. However, it is associated with potential long-term effects such as chronic cardiotoxicity and liver injury (*Wu et al., 2016*; *Joerger, 2016*). The magnitude of anthracycline-induced liver injury (AILI) in breast cancer patients is rising as a result of the increasing number of long-term cancer survivors and because of the tendency to use higher doses of anthracyclines, as well as combined endocrine treatments with synergistic liver toxic effects. As the clinical standard, serum transaminases are generally used for evaluating liver injury (*Kagawa et al., 2018*). However, recent evidence suggests that the release of danger signals may occur in the absence of hepatocyte necrosis and that even minor serum ALT elevations that are often observed in a large percentage of patients treated with drugs reflect hepatocyte death (*Mosedale et al., 2018*). Early diagnosis of AILI is a crucial point for AILI prevention and treatment. Research is needed to understand the mechanisms for AILI, particularly involving its genetic basis.

Currently, exosomes have drawn considerable attention for providing sensitive and detailed insights into many diseases. Exosomes are small extracellular membrane-bound vesicles ranging from 20–150 nm secreted by a variety of cell types, which contain selective particles, including mRNA, miRNA, and proteins. The primary function of these vesicles is cell–cell communication in multiple pathophysiological processes (*Barile & Vassalli, 2017*). MicroRNAs (miRNAs) are small non-coding RNA molecules that can regulate the gene expression post-transcriptionally. With the high stability and protection against RNase-mediated degradation in exosomes, miRNA are attractive candidates to contain valuable biomarkers or exhibit therapeutic effects on experimental and clinical conditions (*Hammond, 2015*). Previous researchers reported that exosome-associated biomarkers

have higher specificity and sensitivity than transaminases, importantly, their serum levels appeared to rise earlier than serum transaminases levels (*Masyuk, Masyuk & LaRusso, 2013*). Therefore, exosomal miRNAs are generally considered stable in peripheral blood and may serve as minimally invasive biomarkers for early detection and prognosis of DILI. Whereas, the role of exosomal miRNAs in AILI remains poorly understood.

In the current study, small RNA sequencing was performed to detect exosomal miRNAs in the postoperative breast cancer patients after receiving anthracycline-based adjuvant chemotherapy and DE-miRNAs were identified between the non-liver injury group and liver injury group. Moreover, bioinformatics analysis revealed the potential functions of DE-miRNAs and recognized miR-1-3p is critical exosomal miRNA and may become a novel biomarker for early detection, diagnosis, and treatment of AILI.

## MATERIALS & METHODS

### Study participants

We recruited 7 postoperative breast cancer patients at the National Cancer Center/Cancer Hospital, Chinese Academy of Medical Sciences (Beijing, China), from September 2017 to January 2018. Patients with any of the following conditions or previous treatments were ineligible: previous invasive breast cancer; non-breast cancer within 5 years before randomization, with the exception of carcinoma in situ of the cervix or colon, melanoma in situ, and skin basal-cell or squamous-cell carcinomas; any previous chemotherapy or radiotherapy for cancer; abnormal renal or hepatic function; and concurrent diseases interfering with planned laboratory tests, such as diabetes, hypertension, etc. Only those participants who received 4 cycles (every 3 weeks) of epirubicin or doxorubicin plus cyclophosphamide, followed 4 cycles (every 2 weeks) of paclitaxel were eligible for this study. The cumulative dose of epirubicin exceeded 320 mg/m$^2$. Written informed consent was obtained from all patients before enrollment, and the study was approved by the institutional ethical committee. National Cancer Center/Cancer Hospital, Chinese Academy of Medical Sciences granted Ethical approval to carry out the study within its facilities (Ethical Application Ref: 17-223/1774).

### ALT detection and grouping of patients

In order to assess the liver function after chemotherapy, approximately 3ml of blood from each patient was collected in SST serum separation tubes (Becton Dickinson) and centrifuged at $3000 \times g$ for 10 min at 4 °C. The serum of all patients was analyzed for ALT using a Cobas c501 analyzer (Roche Diagnostics, Germany), and kits were procured by Roche. The criterion of liver injury was set as ALT>40U/L (*Yu et al., 2017*).

### Sample collection and exosomes extraction

All of the whole blood samples from individuals were obtained and collected in K2EDTA tubes (Becton Dickinson) using standard venipuncture procedures. After centrifugation at 3,000 g for 15 min at 4 °C, the plasma was aspirated into micro tubes and stored at −80 °C fridge before use.

plasma-derived exosomes were isolated by differential centrifugation according to the protocol previously described (*Théry et al., 2006*). After thawing at 37 °C, plasma was

centrifugated at 3,000 g at 4 °C for 15 min to remove cell debris. The supernatant was diluted by a seven-fold volume of phosphate-buffered saline (PBS), centrifuged at 13,000 g at 4 °C for 30 min, and filtered through a 0.22 μm filter to eliminate large particles. Then, the supernatant was ultracentrifuged using a P50A72-986 rotor (CP100NX; Hitachi, Brea, CA, USA) at 100,000 g at 4 °C for 2 h to pellet the exosomes. The pellet was resuspended in PBS and centrifuged again at 100,000 g at 4 °C for 2 h. After PBS washing, exosome pellets were resuspended in 100 μl PBS for TEM, NTA, and further research.

## Exosomes characterize validation

The exosome pellets were observed and photographed by a transmission electron microscope (JEOL-JEM1400, Tokyo, Japan). Briefly, 10 μl exosomes solution was dropped onto a square copper grid and incubated for 10 min at room temperature. After washing with sterile distilled water, the exosomes were negatively stained with uranyl oxalate solution for 1 min. Then, the sample was dried under incandescent light for 2 min before viewing.

For the purpose of measuring size distribution and concentration, nanoparticle tracking analysis (NTA) was performed using the ZetaView PMX 110 (Particle Metrix, Meerbusch, Germany) and corresponding software NTA software (ZetaView 8.02.28) as recommended by the company.

Western blot analysis was performed to detect exosomal surface markers. Briefly, the exosome supernatant was denatured in sodium dodecyl sulfonate (SDS) buffer and then incubated with CD63, TSG101, Alix and Calnexin. The proteins were finally visualized with a Tanon4600 Automatic chemiluminescence image analysis system (Tanon, Shanghai, China).

## MicroRNA sequencing and data analysis

Total RNA in plasma-derived exosomes was extracted and purified using a miRNeasy® Mini kit (Qiagen, Cat. No. 217004) according to the manufacturer's instruction. The degradation and contamination of RNA were monitored on 1.5% agarose gels. The concentration and purity of RNA were assessed using NanoDrop 2000 Spectrophotometer (Thermo Fisher Scientific, Wilmington, DE) and RNA Nano 6000 Assay Kit of the Agilent Bioanalyzer 2100 System (Agilent Technologies, CA, USA). The concentration of RNA is more than 500 pg/ul in all samples. Library preparation and sequencing of miRNA were performed by Beijing ECHO BIOTECH Co. Ltd. In brief, after ligated with 3′ and 5′ adaptors, small RNAs were reverse-transcribed to cDNA and amplified by PCR. The products from PCR were sequenced on an Illumina HiSeq 2500 platform (Data S1). Considering that some miRNAs have little or no expression in some samples, only those with raw count value > 3 in more than 75% of samples were retained for further analysis. The fold-change and *P*-value were calculated based on the miRNA counts using the package edgeR (*Robinson, McCarthy & Smyth, 2010*) of R and differentially expressed miRNAs were identified between groups of interest. *P*-value < 0.05 and |log2FC (fold change) |>1 were set as the cut-off criteria to screen out DE-miRNAs (Code S1).
### Identification of AILI-related target genes

miRTarBase (*Chou et al., 2018*), an experimentally supported microRNA-target interactions database, was applied to predict the targets of the known miRNAs. The target genes of novel miRNAs were predicted using miRDB (*Liu & Wang, 2019*) and the screening criterion was set as score $\geq$80. Besides, we searched "drug-induced liver injury" in the Comparative Toxicogenomics Database (CTD, revision 15982) (*Davis et al., 2019*) and got the DILI-related genes (marker, mechanism or therapeutic genes of DILI). To narrow the research scope for further research, the intersection of predicted genes and DILI-related genes were identified as the AILI-related target genes of the DE-miRNAs.

### Functional annotation and pathway enrichment analysis

Gene Ontology (GO) annotation and Kyoto Encyclopedia of Genes and Genomes (KEGG) pathway enrichment analysis for AILI-related target genes of the DE-miRNAs were performed using the database for annotation, visualization and integrated discovery (DAVID 6.8) (*Huang, Sherman & Lempicki, 2009*). Only the GO terms and KEGG pathways with $P < 0.05$ were considered statistically significant.

### PPI network and miRNA-hub gene network construction

To better understand the interaction among the AILI-related target genes, the protein-protein interaction (PPI) network was generated using the STRING database (*Szklarczyk et al., 2015*) and only the interactions with a combined score $> 0.4$ were considered significant. Then, the PPI network was visualized and analyzed using Cytoscape software (version 3.7.1) (*Shannon et al., 2003*). The Cytohubba plug-in of Cytoscape (version 0.1) (*Chin et al., 2014*) was used to identify important hub genes of the entire network. The maximal clique centrality (MCC) method was used to identifying hub objects in this study because this method could capture more essential proteins in the top-ranked list in both high degree and low degree proteins. After that, the miRNA-hub gene network was constructed by Cytoscape software.

## RESULTS

### Serum biochemical changes in patients

Seven postoperative breast cancer patients who underwent chemotherapy were recruited in our study. To assess liver function, serum ALT level was measured in all individuals before and after chemotherapy. According to the results of the serum ALT level (Fig. 1, Data S1), seven patients were divided into two groups: the non-liver injury group ($n = 2$) and the liver injury group ($n = 5$).

### Verification of exosomes

After isolated from the plasma, exosomes were verified by TEM, NTA and western blot. The morphology of exosomes was visualized under a transmission electron microscope (TEM), and we found homogeneous cup-shaped vesicles with lipid bilayer membranes (Fig. 2A). We then performed a nanoparticle tracking analysis (NTA) to measure the size distribution of exosomes, and the results revealed the most widely distributed particle diameter was

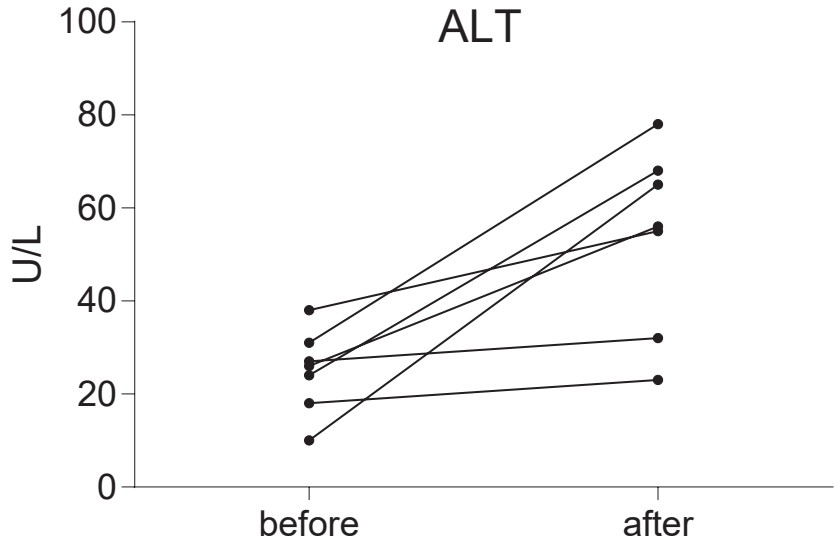

**Figure 1** Before-after plots demonstrate the changes of serum ALT levels in breast cancer patients during chemotherapy. Serum transaminase levels were significantly increased in five of the patients, but not significantly in the other two.

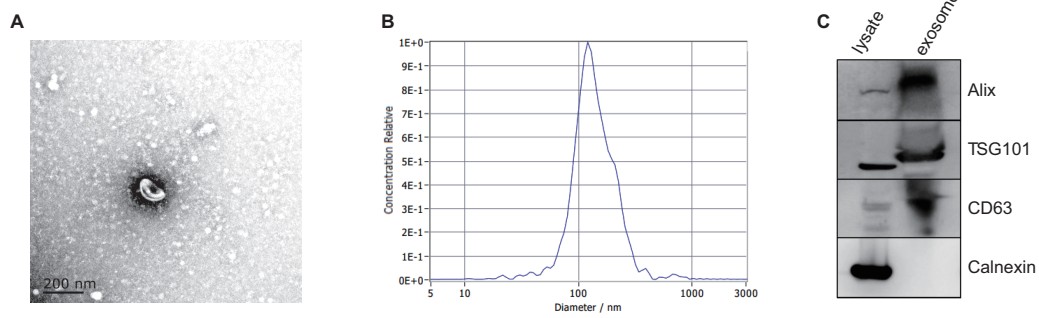

**Figure 2** Characterization of exosomes isolated from plasma. (A) Representative TEM images of exosomes isolated from plasma. Scale Bar = 200 nm. (B) Nanoparticle tracking analysis revealed a size distribution of the exosomes. (C) Western blot analysis for exosomes marker in exosomes and cell lysates.

122.9 nm (Fig. 2B). Additionally, exosomal surface markers were detected using western blot analysis. As shown in Fig. 2C and Data S2, exosome-positive markers (Alix, TSG101, and CD63) were identified, while the negative marker for exosomes (Calnexin) was absent in the isolated exosomes. These results demonstrated that exosomes were successfully isolated from the plasma.

## Identification of DE-miRNAs and AILI-related target genes

We performed high throughput sequencing to detect the miRNA expression of exosomes and screened DE-miRNAs between non-liver injury group and liver injury group using edgeR with $P < 0.05$ and $|\log_2 FC| > 1$. The results showed that a total of 2,031 miRNAs had been detected, including 1576 known miRNAs and 455 novel miRNAs. There are
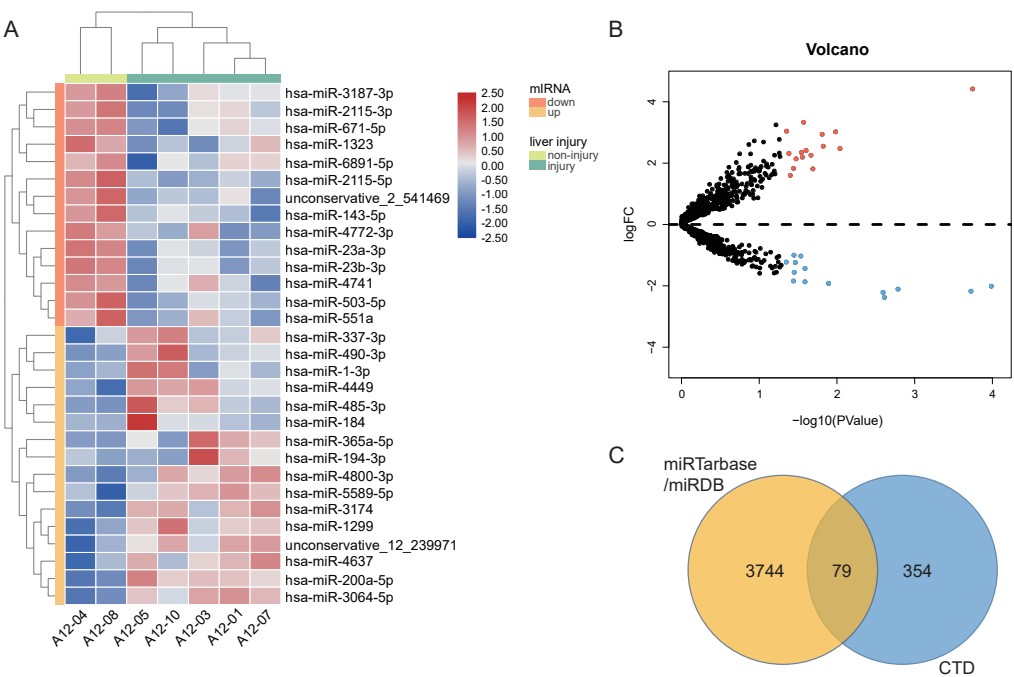

**Figure 3 The different exosomal miRNA expression profiles and screening of AILI-related target genes.** (A) Heat map of DE-miRNAs. Red represents higher expression and blue represents lower expression. (B) Volcano plot of DE-miRNAs. The red and blue points represent significantly upregulated and downregulated miRNAs, respectively. (C) Venn diagram of target genes overlapped with DILI-related genes.

30 DE-miRNAs were screened, of which 16 miRNAs (1 novel miRNA and 15 known miRNAs) were upregulated, and 14 miRNAs (1 novel miRNA and 13 known miRNAs) were downregulated. The volcano plots and heatmap of DE-miRNAs between the two groups are shown in Fig. 3. Then, we employed two miRNA-target interactions database, miRTarBase and miRDB, to predict the target genes of known and novel DE-miRNAs, respectively. As a result, 3823 predicted targets from miRTarBase and miRDB were identified as the target genes of 30 DE-miRNAs. Genes of DILI were downloaded from the CTD Database. A total of 433 DILI-related genes were recorded. To narrow the research scope, the intersection of predicted genes and DILI-related genes were identified as the AILI-related target genes of the DE-miRNAs. Finally, 79 AILI-related target genes were screened, including 55 and 36 targets for upregulated and downregulated DE-miRNAs, respectively (Fig. 3C).

## Functional analysis of the AILI-related target genes

To evaluate the biological functions of these screened AILI-related target genes, GO annotation and KEGG pathway enrichment analysis were performed by using DAVID (Table S1).

Three functional categories were selected in GO annotation, including biological process (BP), cellular component (CC), and molecular function (MF). In the BP category, AILI-related target genes of upregulated miRNAs were significantly enriched in response to drug, positive regulation of MAPK cascade, and response to amino acid. In the CC category, these genes were significantly enriched in extracellular exosome, extracellular space, and cell–cell adherens junction. In addition, these genes were mainly enriched in growth factor activity, cadherin binding involved in cell–cell adhesion, and oxidoreductase activity in the MF category (Fig. 4A). Besides, AILI-related target genes of downregulated miRNAs were significantly enriched in cellular response to vascular endothelial growth factor stimulus, coronary vein morphogenesis, and response to lipopolysaccharide in the BP category. In the CC category, they were mainly enriched in extracellular exosome, myelin sheath, and platelet alpha granule lumen. Moreover, these genes were significantly enriched in enzyme binding, platelet-derived growth factor receptor binding, and chemoattractant activity in the MF category (Fig. 4B).

KEGG pathway analysis was further conducted for AILI-related target genes of DE-miRNAs. The most significantly enriched pathways of upregulated miRNAs included adherens junction, rheumatoid arthritis, and HIF-1 signaling pathway (Fig. 4A). The AILI-related target genes of downregulated miRNAs were enriched in some terms but none of them are significant (Fig. 4B).

### PPI network and miRNA-hub gene network construction

The PPI network of AILI-related target genes was constructed using the STRING database (Fig. 5A). Then, we applied Cytohubba plug-in of Cytoscape to screen out the top 10 hub genes of the network using the MCC method (Table 1). In the PPI network, the hub genes were IL6, VEGFA, CCL2, HMOX1, IGF1, ACTB, CXCL1, SOD2, NOTCH1, and ARG1. Subsequently, the miRNA-hub gene network was constructed by Cytoscape software. As shown in Fig. 5B, the hub target genes are regulated by 6 DE-miRNAs and miR-1-3p regulates the most hub genes($n = 8$). Our results suggest that miR-1-3p may play an important role in AILI.

## DISCUSSION

Recently, researchers and clinicians have found that exosomal miRNAs are more likely to be applied as promising minimally invasive biomarkers in drug-induced liver injury. *Motawi et al. (2018)* reported that exosomal miR-122a-5p exhibited higher diagnostic performance with a broader diagnostic time window and an earlier diagnostic potential than its corresponding total serum level in thioacetamide (TAA)-induced acute liver injury. Similarly, in the absence of overt hepatocellular toxicity, significant elevation of exosomal miR-122 was observed in primary human hepatocytes at subtoxic acetaminophen (APAP) concentration for 24 h (*Holman et al., 2016*).

To our knowledge, this study is the first to investigate the expression profiling of plasma exosomal miRNAs in breast cancer patients with AILI. According to the detection results of alanine aminotransferase in patients after chemotherapy, we found that anthracycline-based adjuvant chemotherapy may induce liver injury. Then, we performed bioinformatics

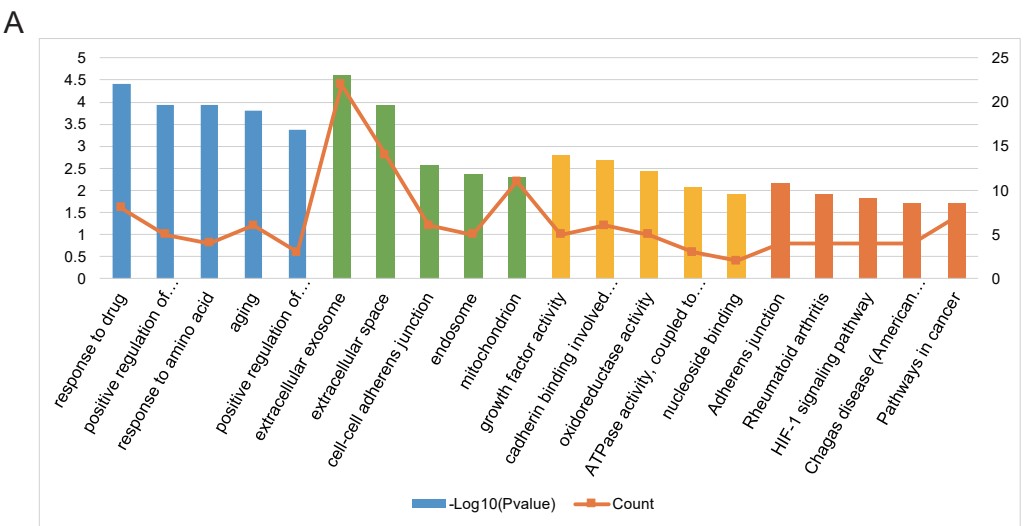

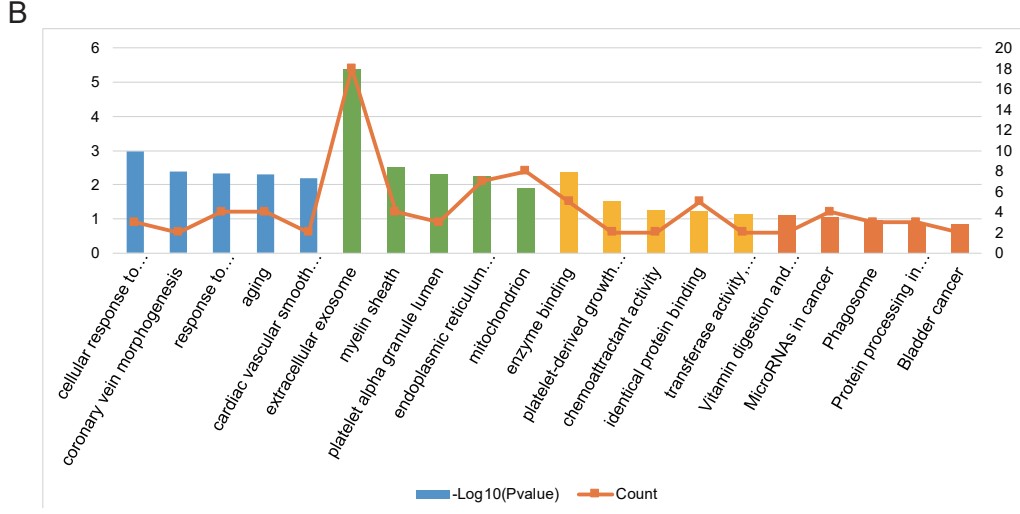

**Figure 4** **Bioinformatics analysis of AILI-related target genes.** (A) GO annotation and KEGG pathway enrichment analysis for AILI-related target genes of the upregulated miRNAs. (B) GO annotation and KEGG pathway enrichment analysis for AILI-related target genes of the downregulated miRNAs. The blue, green, yellow and orange bars represent the enrichment analysis results of BP, CC, MF, and KEGG, respectively.

analysis to investigate the miRNA expression profile in exosomes and found the potential mechanism of AILI.

The KEGG pathway enrichment analysis was performed on the AILI-related target genes. It's worth noting that some terms have been demonstrated to be associated with drug-induced liver injury. FoxO signaling pathway involved in oxidative stress-induced apoptosis and there is evidence that FoxO3 serves a proapoptotic role in hepatocellular apoptosis under oxidative stress (*Tao et al., 2013*). Activated HIF-1 signaling pathway promoted hepatocellular necrosis at the early time of APAP toxicity (*Sparkenbaugh et al.,*

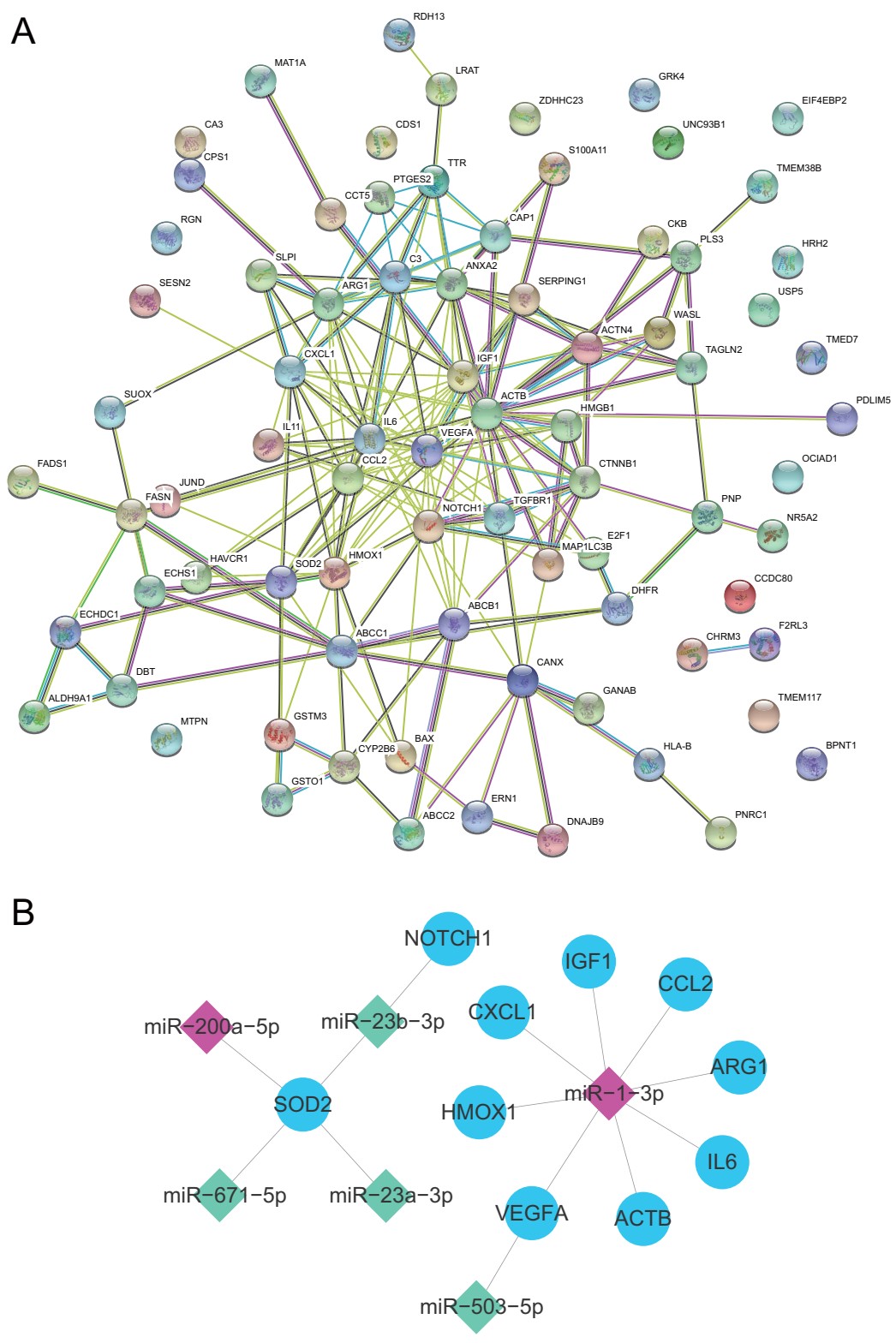

**Figure 5 PPI network and miRNA-hub gene network.** (A) PPI network of AILI-related target genes. (B) miRNA-hub gene network. In the miRNA-hub gene network, the blue ellipses represent hub genes, the red diamonds represent upregulated miRNAs and the green diamonds represent downregulated miRNAs.

**Table 1  Top 10 hub genes in the PPI network (ranked by MCC method).**

| Rank | Name | Score |
|------|------|-------|
| 1 | IL6 | 8670 |
| 2 | VEGFA | 8667 |
| 3 | CCL2 | 8172 |
| 4 | HMOX1 | 7938 |
| 5 | IGF1 | 7626 |
| 6 | ACTB | 6838 |
| 7 | CXCL1 | 6726 |
| 8 | SOD2 | 5052 |
| 9 | NOTCH1 | 1808 |
| 10 | ARG1 | 1748 |

*2011*). In addition, Heatstroke-induced hepatocyte exosomes aggravate liver damage and inflammation by activating the NOD-like receptor signaling pathway in mice, which has been validated in the study of *Li et al. (2019)*. Therefore, these pathways are most likely involved in the pathogenesis and progression of AILI.

Based on the STRING database and Cytohubba plug-in of Cytoscape, significant hub genes have been screened out from the PPI network. Interleukin 6 (IL6), which encodes a cytokine involved in inflammation, is the hub gene with the highest score in the network. Previous studies have shown that IL-6 can promote liver regeneration and repair, but it can also make the liver sensitive to injury, stimulate hepatocyte apoptosis, induce insulin resistance, and participate in the occurrence and development of non-alcoholic steatohepatitis (NASH) (*Braunersreuther et al., 2012*). In the miRNA-hub gene network, superoxide dismutase 2 (SOD2) has the most connections to the DE-miRNAs and three of the four DE-miRNAs interacting with SOD2 are downregulated. SOD2 is an important antioxidant defense against oxidative stress, catalyzing the dismutation of superoxide (O2-) into oxygen and hydrogen peroxide (*Miao & St. Clair, 2009*). *Li et al. (2015)* reported that SOD2 protects the liver from reperfusion injury following severe shock. Therefore, the activation of immune response and the occurrence of oxidative stress may play a key role in the development of AILI.

The miRNA-hub gene network also helps us to find the essential exosomal miRNAs in the development of liver injury. We discovered that most of the hub genes($n = 8$) could be potentially regulated by miR-1-3p. The previous study revealed that miR-1-3p significantly increased in APAP- and TAA-induced hepatocellular injury models. Besides, miR-1-3p also increased in the early stages of α-naphthylisothiocyanate- and 4,4′-methylenedianiline-induced cholestasis models (*Kagawa et al., 2018*). Additionally, researchers have found that the expression level of miR-503-5p was significantly decreased with chronic low-dose exposure to microcystin-LR in mice liver tissues (*Xu et al., 2018*).

There are some limitations to our study. First of all, due to the strict enrollment conditions and the low number of patients with AILI, only seven patients were recruited in this study. It is necessary to recruit more patients to get a more solid conclusion. Secondly, considering that the verification experiment requires a large amount of time, we have not

verified the results of bioinformatics analysis. AILI is highly correlated with patients' quality of life, but there are few studies on it. Through miRNA sequencing and bioinformatics analysis, we got some valuable results and shared them as soon as possible to accelerate AILI research.

## CONCLUSIONS

Taken together, we found that ALT was significantly increased in postoperative breast cancer patients after using the regimens containing anthracycline. Through the assistance of high throughput sequencing and bioinformatics analysis, we identified the potential regulatory functions of differentially expressed exosomal miRNAs and screened out the essential exosomal miRNAs in the development of AILI. Moreover, our study provides an experimental basis for experimental verification to reveal the actual function and mechanism of miRNAs in AILI.

### Funding

This work was supported by the Chinese Academy of Medical Sciences Innovation Fund for Medical Sciences (Grant No.2017-I2M-3-012 & No.2017-I2M-1-013). The funders had no role in study design, data collection and analysis, decision to publish, or preparation of the manuscript.

### Grant Disclosures

The following grant information was disclosed by the authors:
Chinese Academy of Medical Sciences Innovation Fund for Medical Sciences: 2017-I2M-3-012, 2017-I2M-1-013.

### Competing Interests

The authors declare there are no competing interests.

### Author Contributions

- Yue Zhang and Di Wang performed the experiments, analyzed the data, prepared figures and/or tables, and approved the final draft.
- Di Shen analyzed the data, prepared figures and/or tables, and approved the final draft.
- Yang Luo and Yi-Qun Che conceived and designed the experiments, authored or reviewed drafts of the paper, and approved the final draft.

### Ethics

The following information was supplied relating to ethical approvals (i.e., approving body and any reference numbers):
The National Cancer Center/Cancer Hospital, Chinese Academy of Medical Sciences granted ethical approval to carry out the study within its facilities (Ethical Application Ref: 17-223/1774).

## Data Availability

The raw sequence data are available in the Genome Sequence Archive (Genomics, Proteomics & Bioinformatics 2017) in the National Genomics Data Center: CRA002510.

The sequencing data are also available at GEO (GSE148282).

## Supplemental Information

Supplemental information for this article can be found online at http://dx.doi.org/10.7717/peerj.9021#supplemental-information.

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
