# Peer review of "Identification of exosomal miRNAs associated with the anthracycline-induced liver injury in postoperative breast cancer patients by small RNA sequencing"

_PeerJ, doi:10.7717/peerj.9021_

## Round 0.1 · original submission · Major Revisions

As you can see from the attached reports, the invited reviewers raised some concerns about your experimental design and analysis description that you would need further clarifications and discussion.

Reviewer 1 ·

Basic reporting

No comment

Experimental design

No comment

Validity of the findings

Authors reported interesting findings on exosomal miRNAs associated with the
anthracycline-induced liver injury. Seven subjects were recruited and showed significant differences in ALT. To find the potential AILI-related exosomal miRNAs, authors tried to integrate their findings with known DILI-related genes. I have several concerns about this study as following.
Major
1. The study only screened the circulating exosomal miRNAs. No further functional experiments were conducted. So, the authors should state the type of tissues used in the CTD Database and explain the potential regulatory mechanisms.
2. In the method part, authors should demonstrate their criteria or threshold in each step. For examples,
a. in “ALT detection and grouping of patients”, they should state the threshold for grouping and the reasons or references for choosing this threshold;
b. in “MicroRNA sequencing and data analysis”, what is the threshold for purity of RNA?
c. In data analysis part, are the p-values raw values or adjusted values? If adjusted values, which method is used to adjust the p-values?
d. How many exosomal miRNAs were totally identified using small RNA seq?
3. The gene enrichment analysis looks unnecessary for this study. Because the reference dataset is DILI-related genes. All the terms should be just related to DILI.
4. In discussion part, authors only used around five sentences at the end of this part to discuss the major findings in their study. They may consider searching if the previous studies find same miRNAs related to AILI, if the same target genes were found, if the regulatory type (up/down-regulated) was same.
Minor
1. Authors should carefully check the paper again. Like in line 276, “has-miR-503-5p” should be hsa-miR-503-5p.

Reviewer 2 ·

Basic reporting

1. Many abbreviations were not explained in the manuscript. Such as APAP (line 286), NAFLD (line312), etc.
2. In Figure 3C, the number of non-overlapped miRNAs should also be presented.

Experimental design

1. The major concern is that the limited number of patients in the study. Only 7 were recruited: 2 were classified as non-injury and the other 5 were classified as injury. It will significantly lead to cases of bias. More patients need to be recruited to get a more solid conclusion. This limitation should also be discussed in the section of discussion.
2. No experimental evidence were included in the manuscript. Although the author stated that the aim of the work is to identify potential miRNAs, the absence of experimental results makes the work incomplete and less convincing.

Validity of the findings

1. The author should clarify the number of replicates for each sample when performing RNA seq (at least two). The other information should also be included such as the quality of the sequencing and the read depth and length.
2. Line 184, the author should explain the meaning of inference score, and justify the reason to use 100 as the cutoff, considering that this an import step in the analysis.

Additional comments

The manuscript of Zhang et al. has identified differently expressed exosomal miRNA from 7 post chemotherapy patients. They performed small RNA seq and bioinformatics analysis. They suggested that miR-1-3p and miR-503-5p are potential targets for further study. Although the language is clear, several major flaws make it not suitable to be accepted for publication.

---

## Round 0.2 · accepted · Accept

As you can see from the attached reviewer responses, both reviewers were happy with your last changes and I follow their recommendation to accept your manuscript for publication.

Reviewer 1 ·

Basic reporting

no comment

Experimental design

no comment

Validity of the findings

no comment

Additional comments

Generally, authors have answered my questions thoroughly. I think that the design of this study is clear and the finding is meaningful.

Reviewer 2 ·

Basic reporting

Now the manuscript is in a good form to be published.

Experimental design

The issues have been addressed.

Validity of the findings

The findings are substantially improved.

Additional comments

Manuscript of "Identification of exosomal miRNAs associated with the anthracycline-induced liver injury in postoperative breast cancer patients by small RNA sequencing" has been improved substantially over an earlier version.